# The Effects of Dry, Humid and Wear Conditions on the Antimicrobial Efficiency of Triclosan-Containing Surfaces

Abel Guillermo Ríos-Castillo, Carolina Ripolles-Avila and José Juan Rodríguez-Jerez *

Departament de Ciència Animal i dels Aliments, Facultat de Veterinària, Universitat Autònoma de Barcelona, Travessera dels Turons s/n, CP 08193 Barcelona, Spain; abelguillermo.rios@uab.cat (A.G.R.-C.); Carolina.Ripolles@uab.cat (C.R.-A.)

* Correspondence: JoseJuan.Rodriguez@uab.cat; Tel.: +34-935-811-448



**Featured Application: Bacteriostatic protection of surfaces and packaging material.**

**Abstract:** This study evaluated the effects of triclosan-containing polyester surfaces under various conditions at concentrations of between 400 ppm and 850 ppm. *Staphylococcus aureus* was chosen for the tests because it rapidly develops resistance to many antimicrobial agents. The results show that dry and humid conditions have bacteriostatic activity that inhibits the growth of *S. aureus*, with a greater effect under dryness ($p < 0.05$). Further, concentrations as low as 400 ppm showed activities of 0.99 $\log_{10}$ and 0.19 $\log_{10}$ for dry and humid conditions, respectively. The study of the association between triclosan concentrations and bacterial inhibition showed a high correlation for dry ($R^2 = 0.968$) and humid conditions ($R^2 = 0.986$). Under wear conditions, triclosan showed a gradual reduction in its bacteriostatic activity due to successive washing/drying treatments ($p < 0.05$). Thus, the use of triclosan in low concentrations is suggested as achieving bacteriostatic activity. Moreover, its use can be considered as complementary to the cleaning and disinfection procedures carried out in the food industry. However, it must not replace them. Manufacturing processes must be improved to preserve the triclosan properties in the antimicrobial materials to control microorganisms involved in cross-contamination between surfaces and food.

**Keywords:** *Staphylococcus aureus*; triclosan; surfaces; antimicrobial activity; bacterial inhibition; epifluorescence microscopy; humidity; dryness; wear condition

## 1. Introduction

The global rise in public concern about the risks associated with bacterial foodborne diseases has led to the establishment of new government regulations related to food hygiene. The objective of these regulations is to achieve greater protection of the health of consumers. The fulfillment of these regulations has led the health and food processing sectors to adopt measures to establish standardized procedures that guarantee the hygienic conditions of their establishments and products [1–3]. A critical point in these hygiene procedures is the property of microorganisms to adhere, colonize, and form biofilms on all surfaces [4–6]. Furthermore, bacterial adhesion to surfaces results in an increased risk of bacterial cross-contamination to food products and enables the transfer of the pathogens that cause foodborne diseases [7–9]. In addition, non-food contact surfaces such as soils, drains, air conditioning systems, walls, and other environments in the food manufacturing industry allow for the presence of microorganisms and the possibility of forming biofilms [10,11].

Cleaning and disinfection procedures are designed to eliminate microorganisms from surfaces. However, although disinfectant products are effective in reducing microbial contamination, the effects are relatively short-term, and when conditions are adequate for the growth of microorganisms,

they again contaminate surfaces [12–14]. With the primary objective of avoiding bacterial adhesion, methods complementary to cleaning and disinfection procedures have been developed. One method is the use of materials that contain antimicrobial agents that inhibit bacterial growth. Triclosan (2,4,4′-Trichloro-2′-hydroxydiphenyl ether) is a synthetic, nonionic broad-spectrum antimicrobial agent that has been incorporated into a large number of products, such as care products, floors, cutting boards, carpets, and polymeric and packaging materials [15–19]. Although triclosan as an antimicrobial is one of the products that has been widely accepted, high concentrations might have adverse effects, and as such, its use has been limited or restricted [20–24]. Triclosan is also the most studied antimicrobial agent in terms of antimicrobial resistance [25]. A lack of rigor in the formulation of antimicrobial concentrations is one of the main causes of the increase in this agent's resistance [26–28].

*Staphylococcus aureus* is one of the most common potential sources of foodborne disease [29,30]. In the food processing environment, this bacterium is an 'indicator' of deficient hygiene because it can spread from food handlers and food contact surfaces to the entire food chain [31]. *S. aureus* is ubiquitous in the environment because it adapts and survives in adverse conditions for long periods of time. It has the capacity to form biofilms on materials and on food-processing surfaces [32–34]. This bacterium can be even more resistant than Gram-negative bacteria to adverse conditions such as high-pressure treatments and the use of disinfectant products [35–37], and its resistance to antibiotics and antimicrobials is increasing [38–40]. This work aims to provide a better understanding of the effects of environmental conditions on the effectiveness of antimicrobials on surfaces, which are used to prevent bacterial growth. In order to fulfill this objective, this study evaluated the antimicrobial activity of triclosan-containing polyester surfaces in various test concentrations against *S. aureus*. The tests were performed under dry, humid, and after wear conditions.

## 2. Materials and Methods

### 2.1. Bacterial Strain and Test Surfaces

*S. aureus* strain ATCC 6538 was obtained from lyophilized cultures in thermosealed vials (Spanish Type Culture Collection, Universitat de Valencia, Spain). The bacterium was grown in buffered peptone water (BPW) (bioMérieux, Marcy l'Etoile, France) at 37 °C for 24 h. It was then cultivated at 37 °C for 24 h in solid tryptone soy agar (TSA) (Biokar Diagnostics, Beauvais, France) to obtain the bacterial culture for the tests. The test surfaces (2.5 cm$^2$ square) were made of polyester, and triclosan in low concentrations, between 400 and 850 parts per million (ppm), was incorporated into them during the manufacturing, by mixing this agent with all reaction components before the cure system. Prior to using the surfaces, they were cleaned with swabs immersed in 70% isopropyl alcohol (2-propanol) [41]. Three tests were performed based on ISO 22196:2011 [42]: (i) antibacterial activity and (ii) bacterial inhibition tests under dry and humid conditions. There were five tests: one without triclosan used as the control (Id0), and four with triclosan (Id1 Id2, Id3, Id4) at concentrations of between 400 and 850 ppm. (iii) To test the triclosan-containing surfaces under wear conditions, seven samples at concentrations of 400, 450, 500, 600, 700, 800 and 850 ppm were used.

### 2.2. Interfering Substance, Neutralizing Medium and Test Diluent

Bovine albumin was used as an interfering substance to represent real conditions of use [41]. For this purpose, 0.60 g of bovine serum albumin (lyophilized powder ≥96%) (Sigma Aldrich Corp., St. Louis, MO, USA) was dissolved in 100 mL of sterile water. The resulting solution was adjusted to pH 7 ± 0.2 and filtered with a membrane with a maximum pore diameter of 0.45 μm (Millipore Co., Billerica, MA, USA), resulting in a bovine albumin solution with a concentration of 0.3 g/L. The test diluent was composed of 8.5 g of NaCl and 1.0 g of tryptone pancreatic (Oxoid Ltd., Hampshire, Basingstoke, UK) diluted in 1000 mL of sterile distilled water and adjusted to pH 7 ± 0.2. As a medium to neutralize the antimicrobial activity of triclosan during the tests, 30 g of Tween 80® was used (Panreac Química S.A., Barcelona, Spain), diluted in 1000 mL of test diluent.

### 2.3. Procedure to Test the Antimicrobial Activity

*S. aureus* colonies obtained from the bacterial culture for the tests in the stationary phase were transferred to the test diluent and adjusted in a range of $1.5 \times 10^8$ to $5.0 \times 10^8$ colony-forming units/mL (CFU/mL), determined by densitometry (Densimat, bioMérieux, France). Immediately, 1.0 mL of bovine albumin solution was added to 1.0 mL of the adjusted bacterial solution to obtain the bacterial suspension test. Each upper side surface was then inoculated with 50 µL of the bacterial suspension. They were covered with a sterile plastic film of 2.0 cm$^2$ in order to ensure the homogeneous dispersion of the inoculums on each surface. To create humid conditions, the inoculated test surfaces were placed in Petri dishes, which were then placed into a humidified chamber (saturated relative humidity ≥90%) using pieces of paper towels that were moistened with sterile distilled water [42,43]. For the dry condition (relative humidity of 55–65%), the surfaces were introduced into chambers without the moistened paper towel that produced the humidity. For both conditions, the chambers with the inoculated surfaces were covered with platinum film and then incubated at 37 °C for 24 h, according to the standard ISO 22196 [42]. After this time had elapsed, the surfaces were transferred to recipients containing 10 mL of the neutralizing medium and 3.5 g of glass beads of 1.0 mm diameter. The neutralizing recipients with the surfaces were shaken for 1 min and the corresponding dilutions were made. Last, the dilutions were cultivated in TSA agar for 24 h at 37 °C to determine the antimicrobial activity.

### 2.4. Procedure to Test the Bacterial Inhibition Under Dry, Humid and Wear Conditions

The procedures to test the bacterial inhibition of the surfaces under dry, humid and wear conditions were performed with the surfaces that were previously recovered from the recipients with the neutralizing medium employed for the antimicrobial activity test (Section 2.3). The bacterial inhibition tests under the wear condition were performed in three consecutive washing/drying treatments (1st, 2nd, and 3rd treatment). The procedure to test each condition was carried out at room temperature (21 °C ± 2 °C).

For humid and dry conditions, each surface recovered was rinsed with 100 mL of distilled water and then transferred to Petri dishes of 9.0 cm in diameter and containing 10 mL of solid TSA agar. After, 100 µL of sterile distilled water was added to the surfaces of the solid TSA agar. They were then scraped with a pipette tip for 1 min to facilitate the recovery of the viable cells in the distilled water ($<4 \times 10^1$ cells/cm$^2$ recovered), which were then spread over the solid agar using a Drigalsky's loop. The sides of the scraped surface were flipped over and put in contact with the solid TSA agar to enable the diffusion of triclosan. Ten mL of TSA liquid at 45 °C was then poured over the solid agar surrounding the surfaces. Lastly, the Petri dishes were incubated for 24 h at 37 °C.

For the wear condition, in the first treatment, the recovered surfaces were rinsed and sterilized by autoclaving at 121 °C for 15 min. When the surfaces reached room temperature (1 h ± 5 min), they were washed in covered recipients with 200 mL of sterile distilled water under constant agitation at 2500 RPM for 1 min. Thereafter, the surfaces were dried in an airflow oven at 45 °C for 24 h ± 5 min. Each surface was immediately inoculated with a bacterial solution of *S. aureus* at concentrations of between $1.5 \times 10^8$ and $5.0 \times 10^8$ CFU/mL, and then incubated for 24 h at 37 °C. When the incubation time had elapsed, the surfaces were rinsed and transferred to the center of the Petri dishes containing solid TSA agar. For the second and third treatment, the procedure was conducted as in the first treatment, using the surfaces of the previous treatment, previously autoclaved at 121 °C for 15 min. When the surfaces were transferred to the Petri dishes with the solid agar, immediately, in the same way as in the procedure of the bacterial inhibition tests under dry and humid conditions, was added 100 µL of sterile distilled water to each surface, scraped with a pipette tip for 1 min, and then spread using a Drigalsky's loop. The sides of the scraped surface were put in contact with the solid TSA agar and then ten mL of TSA liquid was added, surrounding the surfaces. The surfaces on the petri dishes with TSA agar were incubated for 24 h at 37 °C.

### 2.5. Microscopy

A Live/Dead BacLight™ Bacteria Viability Kit L-13152 (Molecular Probes, Eugene, OR, USA) was used to test the cells resulting from the bacterial inhibition produced by triclosan-containing polyester surfaces by microscopy. The Live/Dead BacLight staining kit, based on the detection of cell membrane integrity by fluorescence, is composed of two nucleic acid-binding stains: SYTO 9 and propidium iodide. The bacterial cells with intact membrane (live cells) are permeable to SYTO 9, but not to propiodium iodide, dying them green. In cells with damaged membrane (dead or injured cells), the two dyes penetrate; however, propidium iodide reduces SYTO 9 producing red fluorescing cells. The surfaces were stained with 30 μL of the staining kit and after 5 min, they were observed by direct epifluorescence microscopy (Olympus® BX51-52, Tokyo, Japan) and coupled with a mercury Olympus® U-RFL-T lamp. Twelve microphotographic images were obtained and processed by each sample with an Olympus DP-50® coupled digital camera and analyzed using the Soft Imaging System® program (AnalySIS® GMBH, Karlsruhe, Germany).

### 2.6. Assessing the Antimicrobial Activity and the Bacterial Inhibition

The bactericidal activity was determined by a reduction of $\geq 2 \log_{10}$ CFU/cm$^2$ from the test surfaces that satisfied the test control validation. The control validation for each test condition was carried out by comparing the control surfaces with those that had triclosan and whose differences did not exceed $0.2 \log_{10}$ [41]. The bacterial inhibition was determined by the average (in cm) measured on the agar from each side of the triclosan-containing surfaces where the absence of bacterial growth was visible. The control surfaces that did not have triclosan in their composition had bacterial growth. The microscopy results of the live (green) and dead or injured (red) bacterial cells of the bacterial inhibition tests were obtained by an average of 10 images for each sample under dry and humid conditions.

### 2.7. Statistical Analysis

Each test was repeated three times and each test material surface was analyzed in triplicate (n = 9). Further, the data were statistically analyzed with SAS® software v 9.1.3.4 (SAS Institute Inc., Cary, NC, USA). The association between triclosan concentrations and the bacterial inhibition under dry, humid, and successive wear conditions was determined by the coefficient of determination ($R^2$), using the mean value results in the range of 0 to 1. The comparison of the dry and humid conditions was performed by the paired two samples Student's t-test. The one-way ANOVA was performed on the data to compare the differences in the antimicrobial activity under dry, humid and wear conditions. Additionally, Duncan's *post hoc* test was used to test the significance of the differences between the antimicrobial activities on the test surfaces ($p \leq 0.05$).

## 3. Results and Discussion

The results of the antibacterial activity ($\log_{10}$ CFU/cm$^2$ reduction) of triclosan-containing polyester surfaces in a concentration range between 400 and 850 ppm under dry and humid conditions against *S. aureus* are shown in Table 1. According to the results, these concentrations demonstrated sufficient activity to inhibit bacterial growth at the lowest used concentration of 400 ppm ($0.99 \log_{10}$ and $0.19 \log_{10}$ reduction for dry and humid conditions, respectively). For the highest concentration of 850 ppm, these reductions were $1.17 \log_{10}$ and $0.20 \log_{10}$. Furthermore, at 500 and 650 ppm, the highest activities of $0.68 \log_{10}$ and $1.02 \log_{10}$, respectively, were observed for dry condition. According to these results, when comparing the concentrations (400, 500, 650 and 850 ppm), in each condition, no significant differences were found in their antibacterial activities ($p > 0.05$). When each concentration was compared between the conditions, statistically significant differences were observed ($p < 0.05$), finding a decrease in the bacteriostatic activity when the samples were tested in humid conditions. However, the activity of triclosan was not very high. In fact, neither the samples preserved in dry or humid conditions achieved a reduction of equal to or greater than $2 \log_{10}$. These results indicate that the antibacterial capacity

of triclosan at the studied concentrations (400 to 850 ppm) is bacteriostatic instead of bactericidal. Therefore, at these concentrations triclosan showed a bacteriostatic activity, and more effective in dry conditions. In our study not bactericidal activity was detected, probably due to the triclosan concentrations selected. Moreover, high humidity was a positive factor for triclosan activity, inhibiting *S. aureus*. Previous studies have shown that in concentrations above 1000 ppm, there is a rapid loss of the activity of triclosan as an antibacterial agent in environmental conditions of refrigeration and in a flood system. Cutter [44] observed that with 1500 ppm of triclosan, they did not have an effect to reduce bacterial populations on refrigerated, vacuum-packaged meat surfaces. In another study, no bactericidal effect was observed with biofilm populations on acrylonitrile-butadiene-styrene (ABS) plastic with 50,000-ppm triclosan, when the tests were performed in a continuous flow of culture reactors with drinking water [45]. According to these studies, an increase in the concentrations of triclosan could be used to observe a bactericidal effect. However, the bacteriostatic or bactericidal activity of triclosan is controversial.

**Table 1.** Bacteriostatic activity (reduction values depicted in $\log_{10}$ CFU/cm$^2$) of polyester surfaces in *S. aureus* treated with triclosan (400 ppm to 850 ppm) and tested under dry and humid conditions.

| Sample | Triclosan (ppm *) | Dry Condition | Humidity Condition |
|---|---|---|---|
| Id1 | 0 | 0.74 ± 0.13 [a,A] | 0.34 ± 0.06 [a,B] |
| Id2 | 400 | 0.99 ± 0.42 [a,A] | 0.19 ± 0.02 [a,B] |
| Id3 | 500 | 0.68 ± 0.16 [a,A] | 0.25 ± 0.03 [a,B] |
| Id4 | 650 | 1.02 ± 0.16 [a,A] | 0.16 ± 0.01 [a,B] |
| Id5 | 850 | 1.17 ± 0.39 [a,A] | 0.20 ± 0.02 [a,B] |

[a,b] Reduction values (means ± s.d.) in the same column with different letters are significantly different ($p < 0.05$). [A,B] Reduction values (means ± s.d.) in the same file (capital letters) with different letters are significantly different ($p < 0.05$). (*) Parts per million concentrations. CFU: colony-forming units.

These results agree with those observed by Møretrø et al. [46], who evaluated the ability of *Staphylococcus aureus*, *Escherichia coli*, *Salmonella*, coagulase-negative staphylococci (CNS) and *Serrratia* spp. to develop and multiply on humid surfaces. In this study, the authors noted that there is a potential danger caused by bacterial cross-contamination from humid surfaces to food. Although the humid condition favored the development of the bacterial cells, the effectiveness of triclosan under this condition also inhibited the growth of *S. aureus*. To this effect, humidity is a determining factor in the survival of bacteria, which can be counteracted by the use of materials with bacteriostatic properties [47–49]. Also, in our study, the use of bovine albumin as an interfering substance to represent real conditions may have had an influence on the action of triclosan, reducing its effect on the surfaces. This fact is in line with Bloomfield et al. [50] and Hunsinger et al. [51], who observed that the presence of bovine albumin produces protective effects, as it interferes with the action of antimicrobials, increasing the survival of microorganisms. Furthermore, bovine albumin may end up acting as a conditioning protein layer, which stimulates a greater adhesion of bacteria and increases their growth in the form of biofilms [52].

The risk of bacterial contamination on dry surfaces is low because bacterial growth and survival are reduced. However, studies in dry conditions show that the viability of *S. aureus* can represent a risk over long periods. In the case of *S. aureus*, it was observed that this bacterium survives in adverse conditions and has the ability to withstand dry environments [7,53]. As such, in air-drying experimental conditions, *S. aureus* represent a risk for at least 72 h [54]. Neely & Maley [55] noted that this bacterium survives more than 90 days on different surfaces. Møretrø et al. [46] showed that surfaces containing triclosan have a better antibacterial protective effect against staphylococci strains in dry conditions than in humid ones. This is significant if we consider that the presence of bacterial pathogens on domestic surfaces, and in the food industry, is a hazard for re-contamination. The drying of the surfaces should not be ruled out after the sanitization process under practical conditions of use because bacterial propagation in a liquid medium is an essential element for the adherence

and possible formation of biofilms. Likewise, a drying surface is a procedure that minimizes the proliferation of microorganisms; however, it has been observed that *S. aureus* can survive in dry conditions and its viability represents a risk over a long period. Therefore, the use of materials with antimicrobial properties should be considered for controlling the spread of this bacterium in different food processing environments.

The objective of antimicrobial agents is to prevent the proliferation of microorganisms and thus, control bacterial adhesion and biofilm formation. This study tested the bacterial inhibition on surfaces to evaluate the concentrations of surfaces that incorporate antimicrobials. Moreover, these surfaces were immersed in TSA agar for 24 h at 37 °C, demonstrating the presence (or not) of a biocidal compound and its concentration (Figure 1). Our results showed that the surfaces used as control (Id1) i.e., not previously treated with triclosan (0 ppm), and tested under dry conditions, formed bacterial colonies of *S. aureus* (Figure 1a). In contrast, the surfaces containing triclosan (Id2) incubated under the humid condition showed bacterial inhibition (Figure 1b).

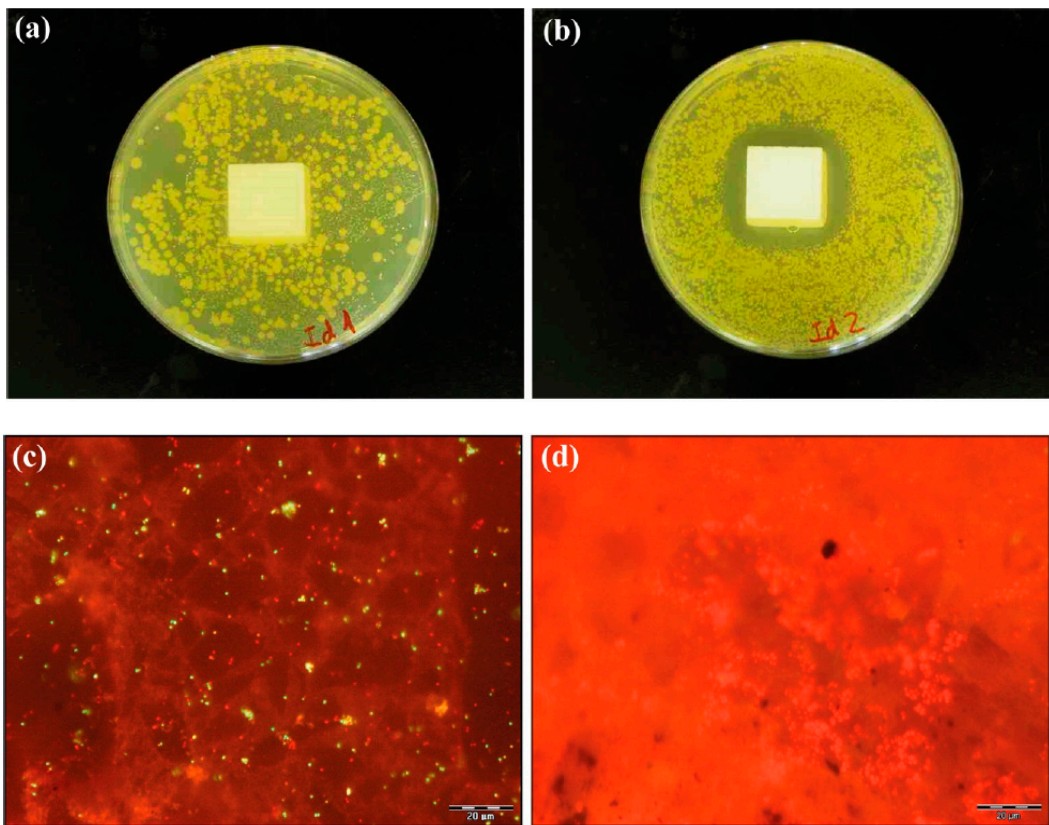

**Figure 1.** Bacterial inhibition on polyester surfaces with the presence of bacterial colonies of *S. aureus* in TSA agar incubated for 24 h at 37 °C: (**a**) bacterial colonies growing on a surface not treated with triclosan tested in dry test conditions; (**b**) bacterial inhibition on a surface treated with 400 ppm of triclosan in humid conditions. Epifluorescence microscopy images of *S. aureus* cells stained with the LIVE/DEAD® kit (Molecular Probes, Eugene, OR, USA) after 24 h incubation: (**c**) presence of *S. aureus* live cells (green) and dead or injured (red) cells on a surface not treated with triclosan; (**d**) inhibition of *S. aureus* growth on a surface treated with 400 ppm of triclosan.

According to the microscopy results (Table 2), there was a development of live bacterial cells on the surfaces used as the control under dry and humid conditions ($1.63 \times 10^5$ and $1.47 \times 10^6$, respectively) (Figure 1c). In comparison with the surfaces used as the control, in the active concentrations (Id2 to Id5), live cells did not develop under dry conditions and showed low counts in Id2 and Id3 under humid conditions ($p < 0.05$) (Figure 1d). Thus, bacterial cells were subjected to high stress by the

presence of triclosan. It was evidenced by the low presence of dead/injured cells for the concentrations at 650 ppm and 850 ppm in dryness ($1.76 \times 10^2$ and $8.42 \times 10^1$, respectively) (Table 2).

**Table 2.** Bacterial inhibition (size of the inhibition is expressed in cm) of *S. aureus* on surfaces treated with triclosan (400 to 850 ppm) or not treated (0 ppm) after 24 h incubation. The presence of live and dead or injured cells by epifluorescence microscopy are expressed in cells/cm$^2$.

| Sample | Triclosan (ppm *) | Dry Condition | | | Humidity Condition | | |
|---|---|---|---|---|---|---|---|
| | | Inhibition Size (cm) | Microscopy (Cells/cm$^2$) | | Inhibition Size (cm) | Microscopy (Cells/cm$^2$) | |
| | | | Live | Dead/Injured | | Live | Dead/Injured |
| Id1 | 0 | - | $1.63 \times 10^5$ | $2.58 \times 10^{4\ a}$ | - | $1.47 \times 10^{6\ a}$ | $2.83 \times 10^{3\ a}$ |
| Id2 | 400 | $0.36 \pm 0.05$ [c,B] | ND | $3.11 \times 10^{1\ b}$ | $0.48 \pm 0.07$ [c,A] | $0.58 \times 10^{1\ b}$ | $1.95 \times 10^{1\ b}$ |
| Id3 | 500 | $0.61 \pm 0.08$ [b,A] | ND | $2.08 \times 10^{2\ b}$ | $0.56 \pm 0.05$ [b,A] | $2.56 \times 10^{1\ b}$ | $0.11 \times 10^{1\ b}$ |
| Id4 | 650 | $0.63 \pm 0.07$ [b,A] | ND | $1.76 \times 10^{2\ b}$ | $0.64 \pm 0.07$ [b,A] | ND | ND |
| Id5 | 850 | $0.88 \pm 0.05$ [a,A] | ND | $8.42 \times 10^{1\ b}$ | $0.95 \pm 0.06$ [a,A] | ND | ND |

[a,b] Values (means ± s.d.) in the same column with different letters are significantly different ($p < 0.05$). [A,B] Values (means ± s.d.) in the same file (capital letters) with different letters are significantly different ($p < 0.05$). (*) Parts per million concentrations. ND: Not detected by epifluorescence microscopy.

The presence of dead or injured cells of *S. aureus* was greater in the control than in the active concentrations ($p < 0.05$). Also, as show in Table 2, the highest bacterial inhibition results were found at the highest concentration (850 ppm). These values were 0.88 cm for the dry condition and 0.94 cm for the humid condition. Although no significant differences were found ($p > 0.05$) between the concentrations of 500 ppm (Id3) and 650 ppm (Id4) in either test conditions, an increase in the size of the inhibition zones was observed. This fact was observed when the relationship between the bacterial inhibition and the concentrations (400, 500, 650 and 850 ppm) was studied for both humid and dry conditions ($R^2 = 0.986$ and $R^2 = 0.968$, respectively) (Figure 2).

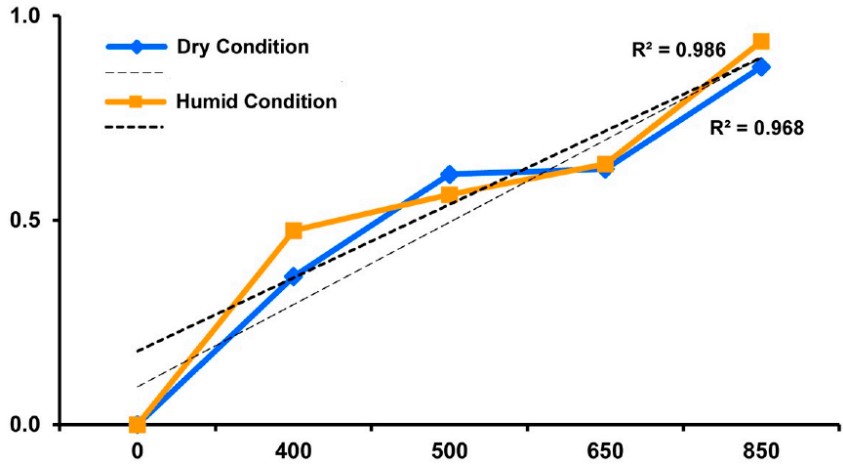

**Figure 2.** Correlation ($R^2$) between triclosan concentration (ppm) and the bacterial inhibition (cm) under dry and humid conditions, tested on the polyester surfaces. N = 9 for each result.

The microscopy and the bacterial inhibition tests also showed the bacteriostatic activity of triclosan. This activity is related to the concentrations in the study. If the concentrations of triclosan had been higher, the biocidal efficacy would also likely have been considerable. However, concentration is equivalent to durability. Greater bacterial inhibition means a greater biocide reserve, causing biological activity to be maintained for a longer period of time. Consequently, the choice of concentration will depend on the desired durability. It has also been indicated that humid conditions are favorable for the colonization and subsequent adherence of bacteria to surfaces. When a biocide is added with bacteriostatic properties, favorable conditions for the formation of biofilms are reduced [49]. In the case of the present study, with concentrations of triclosan between 400 ppm (Id2) and 850 ppm (Id5),

adequate conditions for biofilm formation were not present (Table 2). Additionally, the drying of the surfaces should not be disregarded during the sanitation process because the spread of bacteria in a liquid medium is essential for their survival [56].

Triclosan is incorporated in different materials to provide protection against bacterial growth. However, there is concern about the increased use of triclosan and its possible adverse effects on human health [23,24,57,58]. These effects include endocrine disruption, cytotoxicity, carcinogenic and neurotoxic effects, and reproductive disorders [59,60]. In addition, studies monitored in humans have detected triclosan in urine, blood and breast milk [21]. However, these adverse results on human health after exposure to triclosan are not yet clear [61]. Goodman et al. [62] observed that the health problems attributed to triclosan are not necessarily being produced exclusively by this agent and can also be attributed to other chemical products with similar properties. In real usage, triclosan is not expected to cause health effects when products containing this agent are used according to indications [21]. However, a prudent use of triclosan is recommended [63].

The legislation concerning triclosan has changed over the years in Europe; however, in the US, legislation on the use of this antimicrobial agent is unclear. In Europe, according to a European Decision (2010), triclosan was banned as an additive and its use in any materials in contact with food [20]. In the USA, although the FDA banned the use of triclosan in hand soaps in 2016 [64], this substance continues to be used in other types of products. According to the legislation of this country, the regulation on the use of triclosan has not presented much progress since the first draft in 1974, which was then updated in 1994. However, until now, there has been no final decision on the use of triclosan [19]. However other antimicrobials may be used in materials. The application of the methodology used in this research, may be interesting to be compared with other studies in the future.

Considering this biocide as a prototype for the understanding of other antimicrobial agents, it has also been reported that this substance has increased antimicrobial resistance in laboratory models [28,65,66]. In contrast, with real-life application, the results do not show an increase in antimicrobial resistance after the use of triclosan [20]. Due to its widespread use, triclosan is only partially removed during the wastewater treatment process and it has been found to be persistent in the environment [20,57,67–69]. Based on these references and our results, triclosan should be used on dry surfaces as bacteriostatic, and its use in low concentrations is justified as complementary to cleaning and disinfection procedures, which cannot be overemphasized. This observation could also be transferred to the formulation of the concentrations of use of other agents when the main commercial objective is that they reach high antimicrobial activity.

The results of the wear condition test through successive washing/drying treatments to evaluate the loss of triclosan bacteriostatic properties are shown in Figure 3. The first treatment is shown to be more effective than the second and third treatments ($p < 0.05$) (Figure 3A). Comparing the second and third treatments, differences between 400 and 600 ppm were observed in the activity of triclosan ($p < 0.05$), while in the concentrations between 700 ppm and 850 ppm, no significant losses were observed ($p > 0.05$) (Figure 3A). The highest bacterial inhibition observed in the first and second treatments was for the concentrations of 850 ppm (1.82 cm and 1.18 cm, respectively), while in the third treatment, it was for those of 800 ppm (1.03 cm). The lowest efficacy in the first and second treatment was for the concentrations of 400 ppm (0.69 cm and 0.55 cm, respectively), and in the third treatment, it was for those of 500 ppm (0.33 cm). The relationship between the triclosan concentrations (400 ppm to 850 ppm) and the bacterial inhibition (cm) confirmed a systematic reduction of the triclosan activity from the first to the third treatments. Therefore, the coefficients of determination of each treatment were first ($R^2 = 0.836$), second ($R^2 = 0.857$), and third ($R^2 = 0.819$) (Figure 3B).

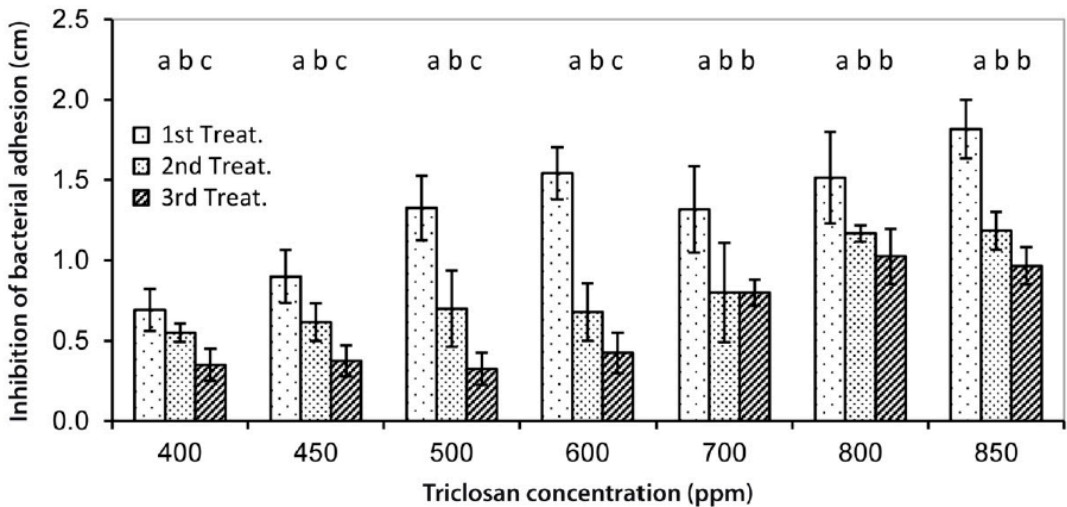

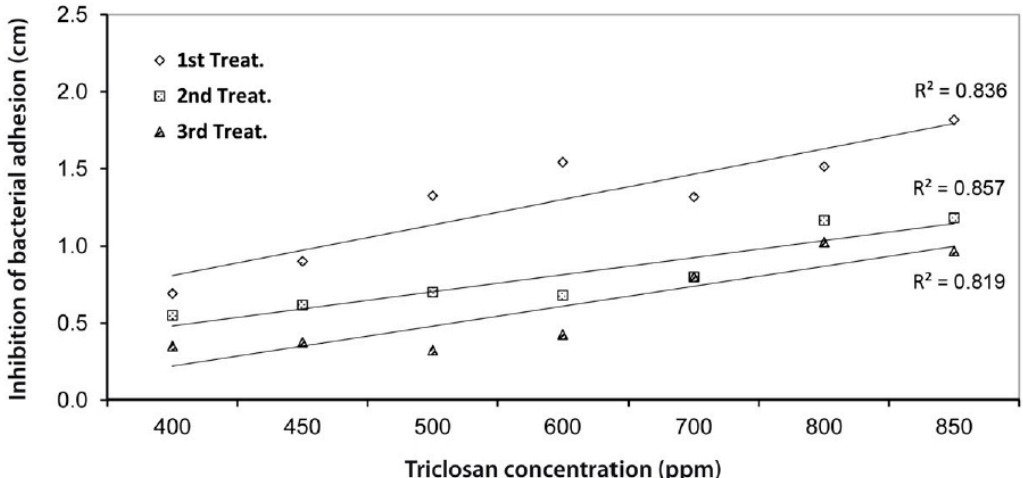

**Figure 3.** (**A**) Inhibition of the bacterial adhesion (cm) under wear condition of *S. aureus* on polyester surfaces treated with triclosan in concentrations between 400 ppm and 850 ppm. The wear condition performed on three successive washing/drying treatments for each concentration. Different letters of mean values over the bars (a, b, c) represent significant differences ($p < 0.05$) between the treatments for each concentration. The error bars represent the standard error of means. (**B**) The relationship between triclosan concentrations and the inhibition of bacterial adhesion for each treatment is represented as the coefficient of determination ($R^2$). N = 9 for each result. Treat: Treatment.

According to the results, which show a decrease in the bacteriostatic effect when triclosan-containing surfaces are subjected to wear conditions, it is expected that they will also lose activity over time, under real-life use. The purpose of incorporating bacteriostatic agents is to ensure adequate hygiene and to inhibit bacterial growth on surfaces, maintaining bacteriostatic properties for long periods. Therefore, dry surfaces will maintain antibacterial properties as a long-lasting effect.

Antibacterial components in appropriate concentrations can be incorporated during the manufacturing process for surfaces that do not come into direct contact with food to ensure more hygienic environmental conditions. This must be as homogeneous as possible, so as not to alter the physical properties of the material used. In the case of compounds such as triclosan, these bacteriostatic properties depend on the concentrations used. In the case of the present study, if high

concentrations are used, this compound will have bactericidal properties [27,70]. Triclosan has been incorporated into different polymers with wide degrees of success to achieve antimicrobial activity [25]. However, the incorporation method of adding triclosan to the packaging material formulation during the extrusion or injection molding, when the polymeric material is being manufactured, has given satisfactory results. As well, triclosan can be incorporated by mixing with thermoplastic polymers that are cooled in a solid [25]. In the case of the incorporation of triclosan by coating on polymeric surfaces, it consists of adding an antimicrobial agent to a coating solution, which is used to cover the surfaces of the packaging material [19]. In order to avoid the possible toxicity of triclosan in the environment, when this agent is added to a surface during its manufacture, studies have been directed towards triclosan grafting by covalent binding to the materials, limiting its release during its usage [63].

Due to current production needs, presently the working time to process food products is frequently increased, with a limited time for sanitation. The incorporation of bacteriostatic agents, such as triclosan, could preserve the hygienic conditions of some materials for a longer time. Thus, antibacterial surfaces can increase the hygienic safety of food industry environments, though their advantages and disadvantages must be measured to obtain a better use in real conditions. Moreover, antimicrobial resistance to antibiotics is becoming more common. As a frequent pathogen, *S. aureus* is transmitted by materials and is resistant to a wide variety of antibiotics, increasing the risk of cross-contamination. The use of triclosan may control *S. aureus* and other bacteria with a high capacity to survive on surfaces.

## 4. Conclusions

This study showed that dryness, humidity and wear conditions have a significant effect on the efficiency of the antimicrobial activity of triclosan. At low concentrations between 400 ppm and 850 ppm, triclosan has bacteriostatic activity under dry and humid conditions. The activity of triclosan in dry conditions was more effective than in humid conditions and bacterial adhesion was prevented. As such, the study shows a high correlation between concentration and bacteriostatic activity. Furthermore, triclosan is sensitive to wear conditions that represent a gradual loss of its bacteriostatic activity. We recommend that the use of triclosan be limited to situations where bacteriostatic protection is required. Furthermore, it should be used in low concentrations because not only may its excessive use have repercussions on human health, but it may also contribute to antimicrobial resistance. Likewise, the manufacturing process must be improved to preserve the triclosan antimicrobial properties in the antimicrobial materials, in particular, to control the microorganisms involved in the direct cross-contamination between surfaces and food.

**Author Contributions:** J.J.R.-J.: designed the experiments, interpreted the results, and contributed to writing the manuscript; C.R.-A.: contributed to writing and reviewing the manuscript; and A.G.R.-C.: executed the experiments, interpreted the results, and wrote the manuscript.

**Funding:** This research received no external funding.

**Acknowledgments:** The authors thank Busquets Soler for her technical assistance in the laboratory and to Sarah Davies for the English grammar review.

**Conflicts of Interest:** The authors declare no conflict of interest.

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
