# Peer review of "The Effects of Dry, Humid and Wear Conditions on the Antimicrobial Efficiency of Triclosan-Containing Surfaces"

_applsci, doi:10.3390/app9081717_

Round 1

Reviewer 1 Report

The aims of the present study at the end of the introduction need rephrasing, because they are not clear.

1. Why was Duncan's chosen as a post-hoc test, instead of for instance Tukey's?

2. Please explain the following statement (line 304) "Nowadays, the period during which food products are manufactured is increasing,..."

3. Describe the concentration range of triclosan usage and discuss why and when it should be used either as bacteriostatic or bactericidal disinfection agent.

4. Please describe and discuss the possible repercussions of triclosan on human health.

5. Please discuss how the manufacturing process must be improved to preserve triclosan properties on food contact surfaces.

Author Response

The aims of the present study at the end of the introduction need rephrasing, because they are not clear.

Response: 

The aims were rewritten for a better understanding (Lines 62 to 66): 

“This work aims to provide a better understanding of the effects of environmental conditions on the effectiveness of antimicrobials on surfaces, which are used to prevent bacterial growth. To fulfill this objective, this study evaluated the antimicrobial activity of triclosan-containing polyester surfaces in various test concentrations against S. aureus. The tests were performed under dry, humid, and after successive wear conditions.” 

1. Why was Duncan's chosen as a post-hoc test, instead of for instance Tukey's?

Response: 

Duncan was used as a post hoctest since there would be averages that could be part of one or another group of concentration treatment. Thus, Duncan gave us greater clarity in the differences between treatment groups of triclosan concentrations. In the case of Tukey, these differences would not have been observed so clearly. 

2. Please explain the following statement (line 304) "Nowadays, the period during which food products are manufactured is increasing,..."

Response: 

This statement refers to the time it takes to process food in the food industry, which is continuous, and therefore, the time for disinfection between each process is limited. Thus, the simultaneous use of surfaces with antimicrobial properties complements the disinfection in the prevention of bacterial growth. Also, the paragraph was rewritten for better understanding. Lines 361 to 363: 

“Nowadays, the increase in working time to process food products is more frequent, which means a more limited time for disinfection. The incorporation of bacteriostatic agents, such as triclosan, could preserve the hygienic conditions of some materials for a longer time”.    

3. Describe the concentration range of triclosan usage and discuss why and when it should be used either as bacteriostatic or bactericidal disinfection agent.

Response: 

The concentration range was added in Lines 73 to 76:

“The test surfaces (2.5 cm2square) were made of polyester, and triclosan in low concentrations, between 400 and 850 parts per million (ppm), was incorporated into them during the manufacturing, by mixing this agent with all reaction components before the cure system”.

Also, added to the Results and Discussion Section in Lines 174 to 182: 

“The results of the antibacterial activity (log10CFU/cm2reduction) of triclosan-containing polyester surfaces in a concentration range between 400 and 850 ppm under dry and humid conditions against S. aureus are shown in Table 1. According to the results, these concentrations demonstrated sufficient activity to inhibit bacterial growth at the lowest used concentration of 400 ppm (0.99 log10and 0.19 log10reduction for dry and humid conditions, respectively). For the highest concentration of 850 ppm, these reductions were 1.17 log10and 0.20 log10. Furthermore, at 500 and 650 ppm, the highest activities of 0.68 log10and 1.02 log10, respectively, were observed for the dry condition. According to these results, when comparing the concentrations (400, 500, 650, and 850 ppm), in each condition, no significant differences were found in their antibacterial activities (p> 0.05).”

In addition, this is discussed in Lines 186 to 199: 

“These results indicate that the antibacterial capacity of triclosan at the studied concentrations (400 to 850 ppm). Therefore, at these concentrations triclosan showed a bacteriostatic activity, and more effective in dry conditions. In our study not bactericidal activity was detected, probably due to the triclosan concentrations selected. Moreover, high humidity was a negative factor for triclosan, inhibiting S. aureus. Previous studies have shown that in concentrations above 1000 ppm, there is a rapid loss of the activity of triclosan as a antibacterial agent in environmental conditions of refrigeration and in a flood system. Cutter [44] observed that with 1500 ppm of triclosan, they did not have an effect to reduce bacterial populations on refrigerated, vacuum-packaged meat surfaces. In another study, no bactericidal effect was observed with biofilm populations on acrylonitrile-butadiene-styrene (ABS) plastic with 50000-ppm triclosan, when the tests were performed in a continuous flow of culture reactors with drinking water [45]. According to these studies, an increase in the concentrations of triclosan could be used to observe a bactericidal effect. However, the bacteriostatic or bactericidal activity of triclosan is controversial.”

4. Please describe and discuss the possible repercussions of triclosan on human health.

Response: 

The discussion about the possible risks on human health was added in Lines 293-300:

“These effects include endocrine disruption, cytotoxicity, carcinogenic and neurotoxic effects, and reproductive disorders [59,60]. In addition, studies monitored in humans have detected triclosan in urine, blood and breast milk [21]. However, these adverse results on human health after exposure to triclosan are not yet clear [61]. Goodman et al. [62] observed that the health problems attributed to triclosan are not necessarily being produced exclusively by this agent and can also be attributed to other chemical products with similar properties. In real usage, triclosan is not expected to cause health effects when products containing this agent are used according to indications [21]. However, a prudent use of triclosan is recommended [63]”

5. Please discuss how the manufacturing process must be improved to preserve triclosan properties on food contact surfaces.

Response:

It was added to the Results and Discussion Section in Lines 351 to 361: 

“Triclosan has been incorporated into different polymers with wide degrees of success to achieve antimicrobial activity [25]. However, the incorporation method of adding triclosan to the packaging material formulation during the extrusion or injection moulding, when the polymeric material is being manufactured, has given satisfactory results. As well, triclosan can be incorporated by mixing with thermoplastic polymers that are cooled in a solid [25]. In the case of the incorporation of triclosan by coating on polymeric surfaces, it consists of adding an antimicrobial agent to a coating solution, which is used to cover the surfaces of the packaging material [19]. In order to avoid the possible toxicity of triclosan in the environment, when this agent is added to a surface during its manufacture, studies have been directed towards triclosan grafting by covalent binding to the materials, limiting its release during its usage [63].”

Reviewer 2 Report

This manuscript evaluated the effects of triclosan-containing polyester surfaces under dry and humid conditions at concentrations of between 400 ppm and 850 ppm against Staphylococcus aureus.The introduction was well- written. However, there are some issues with experimental design and interpretation of the findings. For example, how the bacterial inhibition was measure was unclear and the microscopy images were very unclear and the numbers reported was confusing. Actually no antimicrobial activity was observed. The authors summarized previous studies on similar studies, but did not provide deep enough discussion on the results in this manuscript.

Detailed comments:

1. Line 50: what is the legal status of applying triclosan on food contact surfaces or package materials?

2. Line 72: how was triclosan coated on to the test surfaces?

3. Line 100: what was the humidity under dry condition in the chamber? Since there was 1 ml suspension, how long did it dry out? What was the water activity on the surface in contact with bacteria suspension? If the bacteria were exposed to a high water activity environment, why it was so called “dry” condition?

4. Line 125: What was the recovery rate of the viable cells? Was the recovery rate consistent under low and high inoculum levels?

5. Line 134: Cell injury not only includes membrane damage but damage or dysfunction of other cell contents. Stain cells with damaged membrane would not represent all injured cells.

6. Table 1: what are the counts, log reductions? Please clarify.

7. Table2: how was “live” defined? Would live cells include injured cells? It was doubted that the number of live cells was 0 for all treated samples while there are “dead/injured” ones under dry condition.

8. Fig 1: The photos were unclear and there was no green nor rend color.

There was no fig 2.

9. Fig 3: How would the numbers reflect the bacterial inhibition? It was not described in the manuscript? How was it measured? What was 1st, 2nd and 3rd treatment? In the materials and method, the description was very confusing.

Author Response

This manuscript evaluated the effects of triclosan-containing polyester surfaces under dry and humid conditions at concentrations of between 400 ppm and 850 ppm against Staphylococcus aureus. The introduction was well- written. However, there are some issues with experimental design and interpretation of the findings. For example, how the bacterial inhibition was measure was unclear and the microscopy images were very unclear and the numbers reported was confusing. Actually no antimicrobial activity was observed. The authors summarized previous studies on similar studies, but did not provide deep enough discussion on the results in this manuscript.

The bacterial inhibition procedure in the Materials and Method Section was revised and rewritten. Also, the microscopy results in Table 2 was revised. The antimicrobial activity was bacteriostatic according to the results; rewritten in Lines 186 to 198 for a better understanding: 

“These results indicate that the antibacterial capacity of triclosan is low at the study concentrations (400 to 850 ppm). Therefore, the ability to control bacterial growth on the surfaces was ‘bacteriostatic’ instead of having bactericidal activity and thus, is more effective in dry conditions. Moreover, high humidity was a negative factor for triclosan, inhibiting S. aureus. Previous studies have shown that in concentrations above 1000 ppm, there is a rapid loss of the activity of triclosan as a bactericidal agent in environmental conditions of refrigeration and in a flood system. Cutter et al. [44] observed that with 1500 ppm of triclosan, they did not have an effect to reduce bacterial populations on refrigerated, vacuum-packaged meat surfaces. In another study, no bactericidal effect was observed with biofilm populations on acrylonitrile-butadiene-styrene (ABS) plastic with 50000-ppm triclosan, when the tests were performed in a continuous flow of culture reactors with drinking water [45]. According to these studies, an increase in the concentrations of triclosan to be used as a bactericidal agent is controversial.” 

Additionally, Results and Discussion Section was extended. It is in red font.

Detailed comments:

1. Line 50: what is the legal status of applying triclosan on food contact surfaces or package materials?

Response:

The legal status is discussed in Lines 301 to 310:

“The legislation concerning triclosan has changed over the years in Europe; however, in the US, legislation on the use of this antimicrobial agent is unclear. In Europe, according to a European Decision (2010), triclosan was banned as an additive and its use in any materials in contact with food [20]. In the USA, although the FDA banned the use of triclosan in hand soaps in 2016 [64], this substance continues to be used in other types of products. According to the legislation of this country, the regulation on the use of triclosan has not presented much progress since the first draft in 1974, which was then updated in 1994. However, until now, there has been no final decision on the use of triclosan [19].However other antimicrobials may be used in materials. The application of the methodology used in this research, may be interesting to be compared with other studies in the future.”      

2. Line 72: how was triclosan coated on to the test surfaces?

Response:

Triclosan was added to the polyester surfaces during their manufacturing Lines 73 to 76: 

“The test surfaces (2.5 cm2square) were made of polyester, and triclosan in low concentrations, between 400 and 850 parts per million (ppm), was incorporated into them during the manufacturing, by mixing this agent with all reaction components before the cure system”.

Likewise, the current methods used for triclosan in materials were expanded in the Results and Discussion Section, Lines 353 to 363: 

Triclosan has been incorporated into different polymers with wide degrees of success to achieve antimicrobial activity [25]. However, the incorporation method of adding triclosan to the packaging material formulation during the extrusion or injection moulding, when the polymeric material is being manufactured, has given satisfactory results. As well, triclosan can be incorporated by mixing with thermoplastic polymers that are cooled in a solid [25]. In the case of the incorporation of triclosan by coating on polymeric surfaces, it consists of adding an antimicrobial agent to a coating solution, which is used to cover the surfaces of the packaging material [19]. In order to avoid the possible toxicity of triclosan in the environment, when this agent is added to a surface during its manufacture, studies have been directed towards triclosan grafting by covalent binding to the materials, limiting its release during its usage [63].”

3. Line 100: what was the humidity under dry condition in the chamber? Since there was 1 ml suspension, how long did it dry out? What was the water activity on the surface in contact with bacteria suspension? If the bacteria were exposed to a high water activity environment, why it was so called “dry” condition?

Responses to each question, separately:

What was the humidity under dry condition in the chamber?

The relative humidity in the chambers during the tests for the dry conditions was added to the text, which was less than 65%. Added to Line 102: “relative humidity of 55% - 65%”. 

Since there was 1 ml suspension, how long did it dry out?

The suspension inoculations did not dry completely during the 24 hours that the tests lasted in dry conditions. This was achieved with the plastic films that we added over the inoculums on the surfaces that prevented complete drying. This is found in Lines 97 to 99: 

Each upper side surface was then inoculated with 50 µL of the bacterial suspension. They were covered with a sterile plastic film of 2.0 cm2to ensure the homogeneous dispersion of the inoculums on each surface”.

The step of covering the inoculated suspensions with a plastic film is indicated in the ISO 22196 standard (referenced in the References Section), on which we based the tests. We have also rewritten the paragraph that mentions the 24 hours that this procedure lasts. Lines 103 - 105: 

“For both conditions, the chambers with the inoculated surfaces were covered with platinum film and then incubated at 37ºC for 24 hours, according to the standard ISO 22196 [42].”  

What was the water activity on the surface in contact with bacteria suspension?

The water activity was not measured in our assay. However, considering that a thin layer of inoculum was maintained during the analysis, we expected it was higher than 0.99 in wet conditions. In dry conditions the water activity may be reduced. Under practical conditions, the activity of water in humid conditions represents a high percentage of relative humidity in the antimicrobial surfaces when they are used in a high percentage of relative humidity. For example, surfaces with continuous cleaning washes, places in refrigeration, and flow systems. In the case of dry conditions, they represented surfaces whose objective is that they are of dry use under environmental conditions.

If the bacteria were exposed to a high water activity environment, why it was so called “dry” condition?

The mention of 'dry conditions' is to indicate that the surfaces were subjected to environmental conditions with a relative humidity between 55% to 65% in the chambers. On the other hand, in humid conditions, water was added to the chambers to increase the relative humidity to more than 90%. Lines 99 to 103: “To create humid conditions, the inoculated test surfaces were placed in Petri dishes, which were then placed into a humidified chamber (saturated relative humidity 90%) using pieces of paper towels that were moistened with sterile distilled water [42,43]. For the dry condition (relative humidity: 55% - 65%), the surfaces were introduced into chambers without the moistened paper towel that produced the humidity”.

4. Line 125: What was the recovery rate of the viable cells? Was the recovery rate consistent under low and high inoculum levels?

Response:

Recoveries were not greater than 4 × 101cells/cm2. This value indicates that enough viable bacterial cells came into contact with the antimicrobial agent during the tests. This information was added in the text, Line 119-122: 

After, 100 µL of sterile distilled water was added to the surfaces of the solid TSA agar. They were then scraped with a pipette tip for 1 minute to facilitate the recovery of the viable cells in the distilled water (< 4 x 101cells/cm2recovered), which were then spread over the solid agar using a Drigalsky’s loop.” 

In the case of surfaces not treated with antimicrobials, the recovery rate should not be higher than 6.2 × 101cells/cm2after 24 hours of incubation according to the ISO 22196 standard (this information is not presented in the text of the manuscript).

5. Line 134: Cell injury not only includes membrane damage but damage or dysfunction of other cell contents. Stain cells with damaged membrane would not represent all injured cells.

Response:

We agree with your opinion. Cells with damaged membranes do not always represent all injured cells. Our purpose in microscopic analysis of surfaces was to complement the procedure, which we did based on the ISO 22196 standard. Even in this standard and others, such as the European EN 13697 or the Japanese JIS Z 2801, on chemicals with bactericidal activity on surfaces, are not considered microscopy as a method to evaluate the results of the activity of antimicrobial products. We are of the opinion that the standards of evaluation of antimicrobial products should consider the microscopy method, since the bacterial survival should distinguish 'viable cells' and 'injured cells' (as you mentioned) from those in the viable but non-culturable state (VBNC, Oliver, 2005 "The viable but nonculturable state in bacteria"). This information is not included in the text of the manuscript since the activity of antimicrobial surfaces is carried out from the colony forming units, that are recovered after the tests, as it is indicated by the current standards.

6. Table 1: what are the counts, log reductions? Please clarify.

Response:

The information was added in Table 1 to clarify:

“Bacteriostatic activity (reduction values depicted in log10CFU/cm2) of polyester surfaces in S. aureus treated with triclosan (400 ppm to 850 ppm) and tested under dry and humid conditions.”.

Also in the footnote of Table 1: 

a-bReduction values (means ± s.d.) in the same column with different letters are significantly different (p < 0.05). A - BReduction values (means ± s.d.) in the same file (capital letters) with different letters are significantly different (p < 0.05). (*) Parts per million concentrations”.

7. Table2: how was “live” defined? Would live cells include injured cells? It was doubted that the number of live cells was 0 for all treated samples while there are “dead/injured” ones under dry condition.

Responses to each question:

How was “live” defined?

The definition of live cells and dead or injured cells are defined in base on the detection of cell membrane integrity by fluorescence microscopy and using dyes. 

“Lives cells” are defined as cells with an intact membrane. This was added to the methods in Lines 145-146: “The bacterial cells with intact membrane (live cells) are permeable to SYTO®9, but not to propiodium iodide, dying them green.”

Would live cells include injured cells?

Using Live/Dead BacLight staining kit, live cells do not include injured cells. Injured and dead cells are defined as the cells without an intact membrane and stained in red with propidium. This information was added to the text to clarify. Lines 146 to 147: “In cells with damaged membrane (dead or injured cells), the two dyes penetrate; however, propidium iodide reduces SYTO®9 producing red fluorescing cells.”  

It was doubted that the number of live cells was 0 for all treated samples while there are “dead/injured” ones under dry condition.

To avoid confusion, because the results were obtained by twelve microscopy images for each sample, we have defined it as (ND) “not detected” by epifluorescence microscopy. This information was added to the results and the footnote of Table 2. The explain of the presence of dead/injured cells, is that the activity of triclosan under dry conditions was more active than in humid conditions. It is described in Lines 246 to 249:

“Thus, bacterial cells were subjected to high stress by the presence of triclosan. It was evidenced by the low presence of dead/injured cells for the concentrations at 650 ppm and 850 ppm in dryness (1.76 x 102and 8.42 x 101, respectively) (Table 2).”

8. Fig 1: The photos were unclear and there was no green nor rend color.

Response:

We have improved Figure 1 using the Soft Imaging System® program (described in Line 151-152). Furthermore, we are attaching the improved Figure 1 to the journal.

There was no fig 2.

Response:

Figure 2 was next to Figure 1. We have separated them.  

9. Fig 3: How would the numbers reflect the bacterial inhibition? It was not described in the manuscript? How was it measured? What was 1st, 2nd and 3rd treatment? In the materials and method, the description was very confusing. 

Responses are presented separately for each question:

How would the numbers reflect the bacterial inhibition?

Response:

The numbers (1st, 2nd, and 3rd) indicate the bacterial inhibition (cm) for the three consecutive washing/drying treatments of the wear condition. The three treatments for each concentration (400, 450, 500, 600, 700, 800 and 850) are shown separately in Figure 3A. Additionally, we improved it in the Material and Methods Section, Lines 113 to 116: “The bacterial inhibition tests under the wear condition were performed in three consecutive washing/drying treatments (1st, 2do, and 3rd treatment).”

How was it measured?

Response:

The bacterial inhibition was measured in centimeters (cm).

What was 1st, 2nd and 3rd treatment?

Response:

Yes, there were three washing/drying treatments of the wear condition. This is also indicated in Figure 3.

In the materials and method, the description was very confusing. 

Response:

The procedure for the humid and dry conditions, as well as the wear condition, were revised and described separately for a better understanding in Lines 117 to 124 for the dry and humid conditions and Lines 125 to 139 for the wear condition. 

Reviewer 3 Report

The submitted manuscript is about the antimicrobial efficiency of triclosan–containing surfaces and how it can be affected by dry, humid and wear conditions. The aim of this study is clearly defined and appropriately framed.

Cleaning and disinfection procedures are critical during food production in order to enhance food safety. The study meets up with important concerns of the food industry such as bacterial adhesion and biofilm formation on surfaces.

The Introduction and the description of the state of the art are clearly articulated.  

Methodology is robust and the experiments are carefully designed and performed. The use of an interfering substance, neutralizing medium and the choice of Staphylococcus aureus as indicator shows experience and expertise of the authors in the subject. Real life use of cleaning and disinfection products is a much more complicated issue and the effectiveness of such products is often limited.                                                                                                        

The Results of the study are sufficiently analyzed and interpreted in a comprehensible way. Furthermore, authors’ conclusions are adequately supported by the results and are clearly related to the literature, which is important in order to help the reader determine what is to be learned and what can be speculative.

The manuscript is well written concerning the English grammar, style and syntax. Some indicative corrections:

Line 146: “…composition had bacteril growth.” should be changed to “…composition had bacterial growth.”

Line 157: “As wel,…”  should be changed to “As well,…”

Public health concerns about the use of triclosan in the food industry are not pinpointed sufficiently in the study (lines 49-51, lines 258-259 and lines 319-320), at least in my opinion. This antimicrobial is banned for use in the manufacture of plastics intended to come into contact with food in many countries. Legal aspects concerning the use of triclosan should be highlighted in a more commentary way to help the reader of the manuscript understand restrictions in use.

Author Response

The submitted manuscript is about the antimicrobial efficiency of triclosan–containing surfaces and how it can be affected by dry, humid and wear conditions. The aim of this study is clearly defined and appropriately framed.

Cleaning and disinfection procedures are critical during food production in order to enhance food safety. The study meets up with important concerns of the food industry such as bacterial adhesion and biofilm formation on surfaces.

The Introduction and the description of the state of the art are clearly articulated.  

Methodology is robust and the experiments are carefully designed and performed. The use of an interfering substance, neutralizing medium and the choice of Staphylococcus aureus as indicator shows experience and expertise of the authors in the subject. Real life use of cleaning and disinfection products is a much more complicated issue and the effectiveness of such products is often limited.                                                                                                       

The Results of the study are sufficiently analyzed and interpreted in a comprehensible way. Furthermore, authors’ conclusions are adequately supported by the results and are clearly related to the literature, which is important in order to help the reader determine what is to be learned and what can be speculative.

The manuscript is well written concerning the English grammar, style and syntax. Some indicative corrections:

Line 146: “…composition had bacteril growth.” should be changed to “…composition had bacterial growth.”

Response: 

It was changed in Line 159.

Line 157: “As wel,…”  should be changed to “As well,…”

Response:

It was changed in Line 171. 

Public health concerns about the use of triclosan in the food industry are not pinpointed sufficiently in the study (lines 49-51, lines 258-259 and lines 319-320), at least in my opinion. 

Response:

Public health concerns were discussed and added to the manuscript in Line 293 to 300: 

“However, there is concern about the increased use of triclosan and its possible adverse effects on human health [23,24,57,58]. These effects include endocrine disruption, cytotoxicity, carcinogenic and neurotoxic effects, and reproductive disorders [59,60]. In addition, studies monitored in humans have detected triclosan in urine, blood and breast milk [21]. However, these adverse results on human health after exposure to triclosan are not yet clear [61]. Goodman et al. [62] observed that the health problems attributed to triclosan are not necessarily being produced exclusively by this agent and can also be attributed to other chemical products with similar properties. In real usage, triclosan is not expected to cause health effects when products containing this agent are used according to indications [21]. However, a prudent use of triclosan is recommended [63].”

This antimicrobial is banned for use in the manufacture of plastics intended to come into contact with food in many countries. Legal aspects concerning the use of triclosan should be highlighted in a more commentary way to help the reader of the manuscript understand restrictions in use.

Response: 

It was added in Lines 301 to 310: 

“The legislation concerning triclosan has changed over the years in Europe; however, in the US, legislation on the use of this antimicrobial agent is unclear. In Europe, according to a European Decision (2010), triclosan was banned as an additive and its use in any materials in contact with food [20]. In the USA, although the FDA banned the use of triclosan in hand soaps in 2016 [64], this substance continues to be used in other types of products. According to the legislation of this country, the regulation on the use of triclosan has not presented much progress since the first draft in 1974, which was then updated in 1994. However, until now, there has been no final decision on the use of triclosan [19].”      
